

# Genome assembly composition of the String "ACGT" array: a review of data structure accuracy and performance challenges

Sherif Magdy Mohamed Abdelaziz Barakat[1], Roselina Sallehuddin[1],
Siti Sophiayati Yuhaniz[2], Raja Farhana R. Khairuddin[3] and Yasir Mahmood[4]

[1] Computer Science, School of Computing, Faculty of Engineering, Universiti Teknologi Malaysia, Skudai, Johor, Malaysia
[2] Advanced Informatics Department, Razak Faculty of Technology and Informatics, Universiti Teknologi Malaysia, Kuala Lumpur, Kuala Lumpur, Malaysia
[3] Department of Biology, Universiti Pendidikan Sultan Idris Tanjung, Malim, Malim, Malaysia
[4] Faculty of Information Technology, The University of Lahore, Lahore, Lahore, Pakistan

Corresponding author
Sherif Magdy Mohamed
Abdelaziz Barakat,
sherifmagdy.barakat@gmail.com

## ABSTRACT

**Background**. The development of sequencing technology increases the number of genomes being sequenced. However, obtaining a quality genome sequence remains a challenge in genome assembly by assembling a massive number of short strings (reads) with the presence of repetitive sequences (repeats). Computer algorithms for genome assembly construct the entire genome from reads in two approaches. The *de novo* approach concatenates the reads based on the exact match between their suffix-prefix (overlapping). Reference-guided approach orders the reads based on their offsets in a well-known reference genome (reads alignment). The presence of repeats extends the technical ambiguity, making the algorithm unable to distinguish the reads resulting in misassembly and affecting the assembly approach accuracy. On the other hand, the massive number of reads causes a big assembly performance challenge.

**Method**. The repeat identification method was introduced for misassembly by prior identification of repetitive sequences, creating a repeat knowledge base to reduce ambiguity during the assembly process, thus enhancing the accuracy of the assembled genome. Also, hybridization between assembly approaches resulted in a lower misassembly degree with the aid of the reference genome. The assembly performance is optimized through data structure indexing and parallelization. This article's primary aim and contribution are to support the researchers through an extensive review to ease other researchers' search for genome assembly studies. The study also, highlighted the most recent developments and limitations in genome assembly accuracy and performance optimization.

**Results**. Our findings show the limitations of the repeat identification methods available, which only allow to detect of specific lengths of the repeat, and may not perform well when various types of repeats are present in a genome. We also found that most of the hybrid assembly approaches, either starting with *de novo* or reference-guided, have some limitations in handling repetitive sequences as it is more computationally costly and time intensive. Although the hybrid approach was found to outperform individual assembly approaches, optimizing its performance remains

a challenge. Also, the usage of parallelization in overlapping and reads alignment for genome assembly is yet to be fully implemented in the hybrid assembly approach. **Conclusion**. We suggest combining multiple repeat identification methods to enhance the accuracy of identifying the repeats as an initial step to the hybrid assembly approach and combining genome indexing with parallelization for better optimization of its performance.

# INTRODUCTION

## Repeats in the genome

Deoxyribonucleic acid (DNA) is the genetic material of most organisms, which is made up of a chain of four chemical bases indicated by letters A, C, G, and T, and a complete set of DNA sequences an organism called genome (*Baxevanis, 2020*). Repetitive sequences were found across all kingdoms of life. More than 50% of the human genome is occupied by DNA repetitive sequences. Repeat is a segment of sequence that appears multiple times in the genome in the identical or near-identical form (*Jain et al., 2018b*; *Venuto & Bourque, 2018*). The essential repeat categories in biology are transposable elements (TE), and tandem repeats (TR) (*Zeng et al., 2018*). TEs are DNA sequences that are able to copy themselves from one genome region, overwriting another region, which are considered to play a major role in the genome evolution, thus, can change the structure and size of genomes, as shown in Fig. 1. The length of TE can be varied, such as long-terminal repeat retrotransposons, the length of which generally ranges from 100 bp to 25 kb (*Liao et al., 2021*). On the other hand, TR is a repetitive DNA sequence, which is either known as a microsatellite for a short repeat segment (1–12 bp) or a minisatellite for a longer repeat segment (12–500 bp) located adjacent to each other. Both types of repeats have the ability to expand their copy number and change the structure and size of the genome (*Paulson, 2018*; *Genovese et al., 2018*). The expansion of repeat sequences can cause many diseases in humans, such as Huntington's disease and Kennedy's disease (*Pinto et al., 2017*). The presence of the repetitive offers crucial biological information that stores historical genome changes within and between species, which should be properly handled during assembly, and cannot be treated as a duplication and then truncated by data cleaning.

## Genome assembly in genome analysis project

Genome analysis can be divided into three main phases; genome sequencing, genome assembly, and genome annotation. In the first phase, genomic sequences from an organism will be extracted and further processed through a sequencing machine, in which the sequences will be fragmented into smaller fragments as an output (reads). In the genome assembly phase, all the reads will be overlapped or mapped to the existing reference genome to construct the genome of the organism. Finally, the assembled genome will

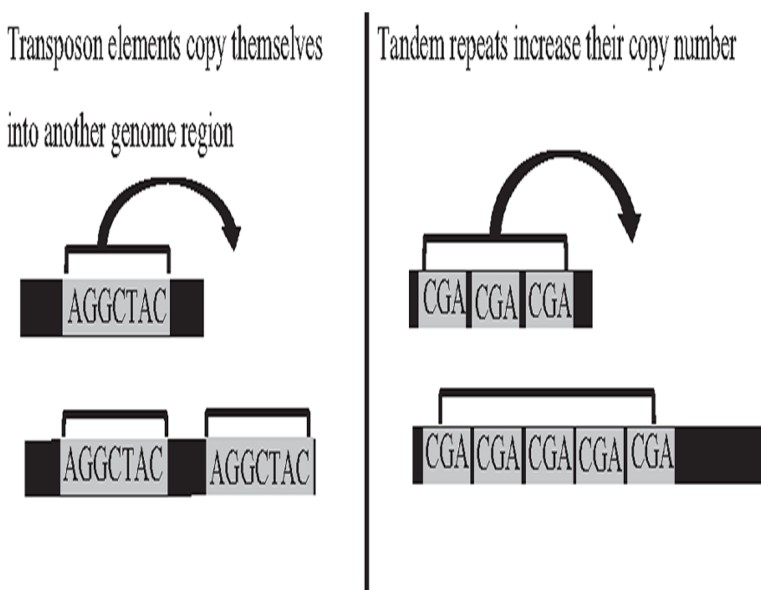

**Figure 1** **Biological activities of TE and TR change genome size.**

be annotated according to their genome position for coding (*e.g.* gene) and non-coding (*e.g.* intron) regions. The genome assembly phase is the most computationally intensive part of genome analysis. Given such a collection of reads, a genome containing all genetic information about an organism will be constructed. In the genome assembly, reads(substrings) are concatenated together to construct the genome. The next generation sequencing (NGS), which is one of the sequencing technology, output (reads) genome reads are available in public repositories such as the National Centre for Biotechnology Information (NCBI) (*Ekblom & Wolf, 2014*), the DNA Databank of Japan (DDBJ), and the UC Santa Cruz Genomics Institute (UCSC) (*Kulkarni & Frommolt, 2017*). Genome data reads can be stored in FASTA and FASTQ files (*Ekblom & Wolf, 2014*).

## Data structure approaches for genome assembly

An exact match is considered when two matched strings are identical, as shown in Fig. 2A. On the other hand, the approximate match represents the high similarity between two strings and is not necessarily identical, and some differences might be there. The number of different letters between similar approximate matched strings is called distance (*Patil, Toshniwal & Garg, 2013*). The distance can be a substitution representing a letter in the target string differently from its corresponding one in the reference string (*Röhling et al., 2020*). The distance can also be an insertion of a letter in the target string that does not exist in its corresponding offset in the reference string. Another type of distance is deletion, which one letter is absent in the target string compared to its corresponding offset in the reference string, as shown in Fig. 2B. *de novo* is Latin word that means from scratch or new. One of the genome assembly approaches is *de novo* approach, which concatenates overlapping reads with the exact match between the suffix of a read with a prefix of another

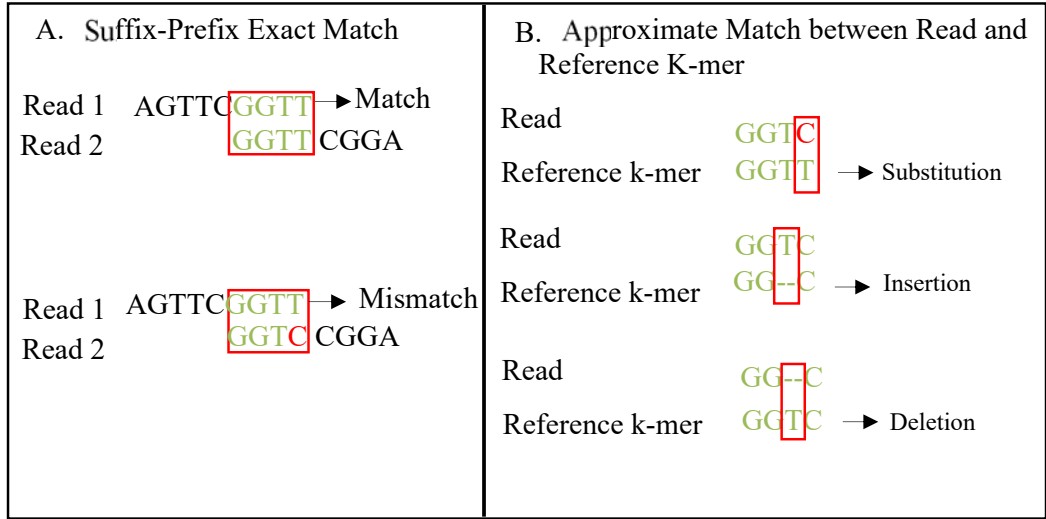

**Figure 2** **String exact and approximate match.**

read (*Baichoo & Ouzounis, 2017*). Overlapping reads will be concatenated into a longer string called a contig. Similarly, overlapping contigs will be concatenated into a scaffold, as shown in Fig. 3 (*Gopinath et al., 2018*). Overlapping distance (length) is a significant parameter for any overlapping algorithm. Overlapping errors have a small impact when sufficient distance is used, as shown in Fig. 4, which suggests that empirical overlapping distance should be greater than 40% from the read length (*Haj Rachid, 2017*).

On the other hand, the assembly is guided by known reference genome instances in the reference-guided approach. Reads are mapped against the reference genome to determine their orders, called the reads alignment process (*Kim, Ji & Yi, 2020*), as shown in Fig. 5. The reads alignment against the reference genome is based on an approximate match between the read and its corresponding k-mer of the reference genome. *Mer* is a Latin word that means part, while k is the length of this part (*Simpson & Pop, 2015*). It is not necessarily the reads exactly match the reference k-mer, because reads of the target genome are slightly different from the used reference genome due to individual biological variation (*Zeng et al., 2018*).

## Limitations of current sequencing technology as the root cause of genome assembly challenges

In 1965 Fred Sanger and colleagues sequenced the first DNA reads to produce reads slightly less than one kilo-base (kb) in length. However, it was first-generation DNA sequencing. Second-generation DNA sequencing or next-generation sequencing (NGS) is a massive number of tiny Sanger sequencing running in parallel. In NGS, large quantities of DNA can be sequenced quickly. Short-length reads are considered the first limitation in this technology, generating a big challenge for assembling repetitive sequences from short reads. Recently, third-generation DNA sequencing (TGS) was introduced by Stephen Quake. The main characteristic of this method is generating longer-length reads. TGS is

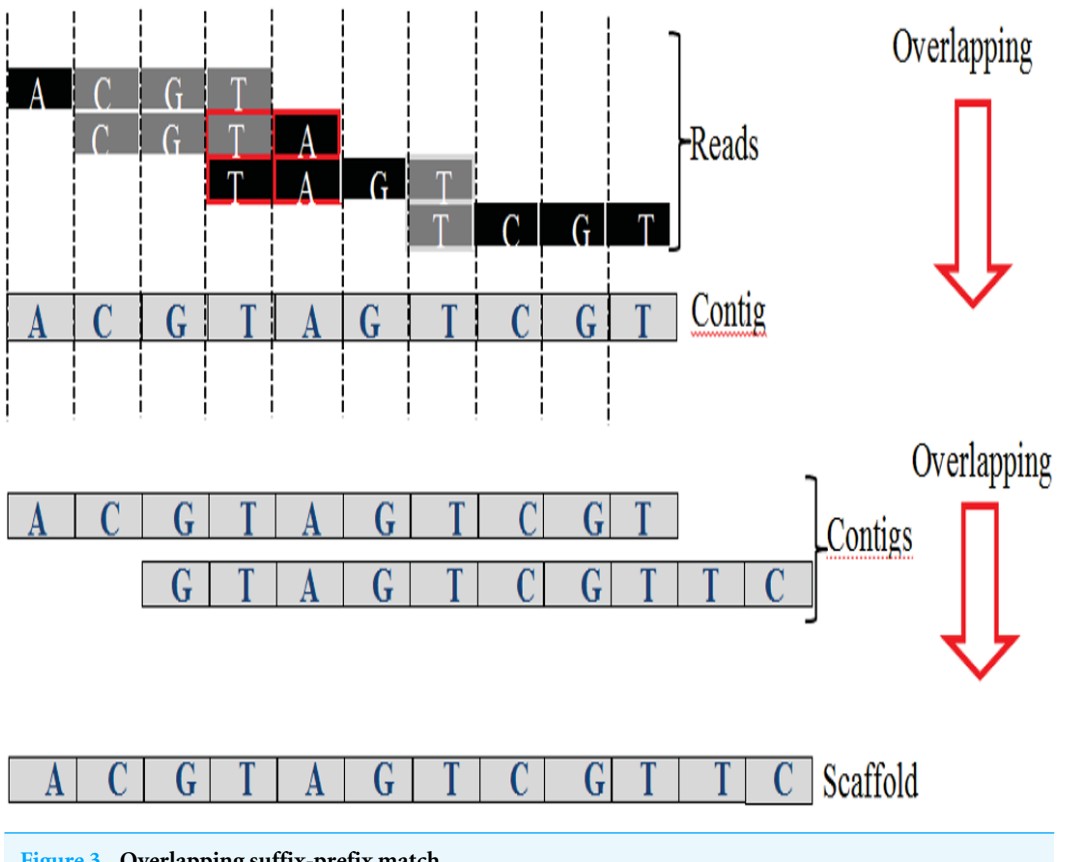

**Figure 3** **Overlapping suffix-prefix match.**

currently available, generating longer-length reads able to bridge long repetitive sequences, and used to develop assemblies such as Canu and Flye (*Guiglielmoni et al., 2021*). However, the natural high error rate of up to 15% of long reads impacts the accuracy of the assembly requiring higher computational complexity correction processes when processing large datasets (*Shafin et al., 2020*; *Liao et al., 2021*). The most common question is why DNA molecules cannot be sequenced as a single string. Firstly, the enzyme used in the sequencing experiment naturally generates sequence errors after sequencing three kilo-base (*Garibyan & Avashia, 2013*). Secondly, the DNA sequencing experiment is too much time-consuming. Thus, sequencing the entire genome using a single sequencing machine would take years to be done (*Angeleska, Kleessen & Nikoloski, 2014*). There is no sequencing platform that provides complete coverage of the whole genome as a single string (*Jain et al., 2018a*). In NGS, each region is sequenced many times in order to ensure that genome is accurately sequenced, called coverage. Data coverage is the average number of times a base of a genome is sequenced to ensure that the genome is sequenced accurately. It is often expressed as (1x, 2x, 3x,…, nx) as shown in Fig. 6. Target genome reads from NGS are extremely massive data. For example, the number of 60X sequence coverage is often mentioned in that the number of letters in the dataset is 60 times bigger than the original genome size (*Jain et al.,*

Given Genome (G): ACTTCACGTCGTCGTCGTTGATCAA

*Long Overlapping Length*

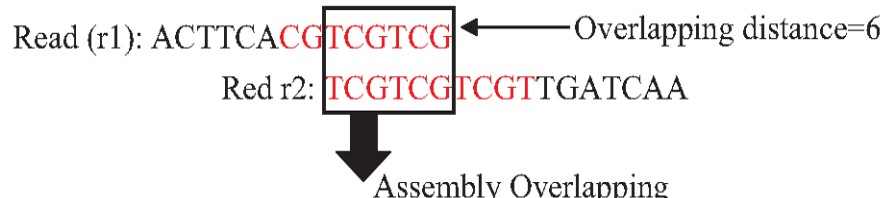

Assembly Overlapping

Genome G: ACTTCACGTCGTCGTCGTTGATCAA    **Correct Genome**

*Short Overlapping Distance*

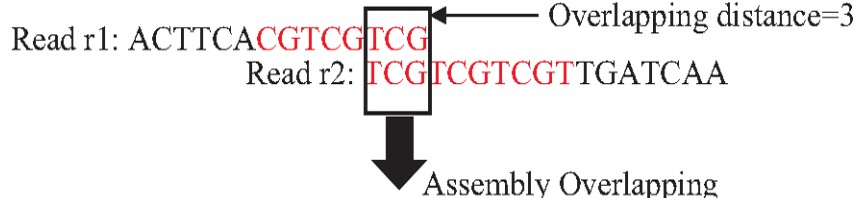

Assembly Overlapping

Genome G: ACTTCACGTCGTCGTCGTCGTCGTTGATCAA
**Wrong Genome (Mis-assembly)**

**Figure 4** **The impaction of overlapping length on assembly accuracy.**

*2018b*). This massive data amount is considered the second limitation of NGS, resulting in big performance challenges during genome assembly.

## Genome assembly challenges
### Assembling repeats from short reads misassembly

The presence of repeats in a genome has increased the complexity of accurately assembl the genome from short reads (*Lohmann & Klein, 2014*; *Acuña Amador et al., 2018*). In the *de novo* approach based on overlapping, algorithms might concatenate reads wrongly, leading to a misassembly simply because they have the same matched overlapping distance (*Simpson & Pop, 2015*; *Wang et al., 2021*). The formulation in Fig. 7 assumes that the given genome G ={TGGGACTGG}, then the genome G is sequenced, resulting in an array of reads R ={GACT, ACTGG, TGGGAC }. According to computer science, concatenating these reads based on overlapping is correct if and only if the reads have an exact suffix-prefix match at a specific overlapping distance. Thus, so both assembled genomes {GACTGGGAC} and {TGGGACTGG} are correct solutions accordingly. However, because of the ambiguity that is resulted from repeats in short reads, when the genome is assembled genome to be {GACTGGGAC} while the real one is {TGGGACTGG}, this problem is a well-known problem called misassembly (*Medvedev, 2019*).

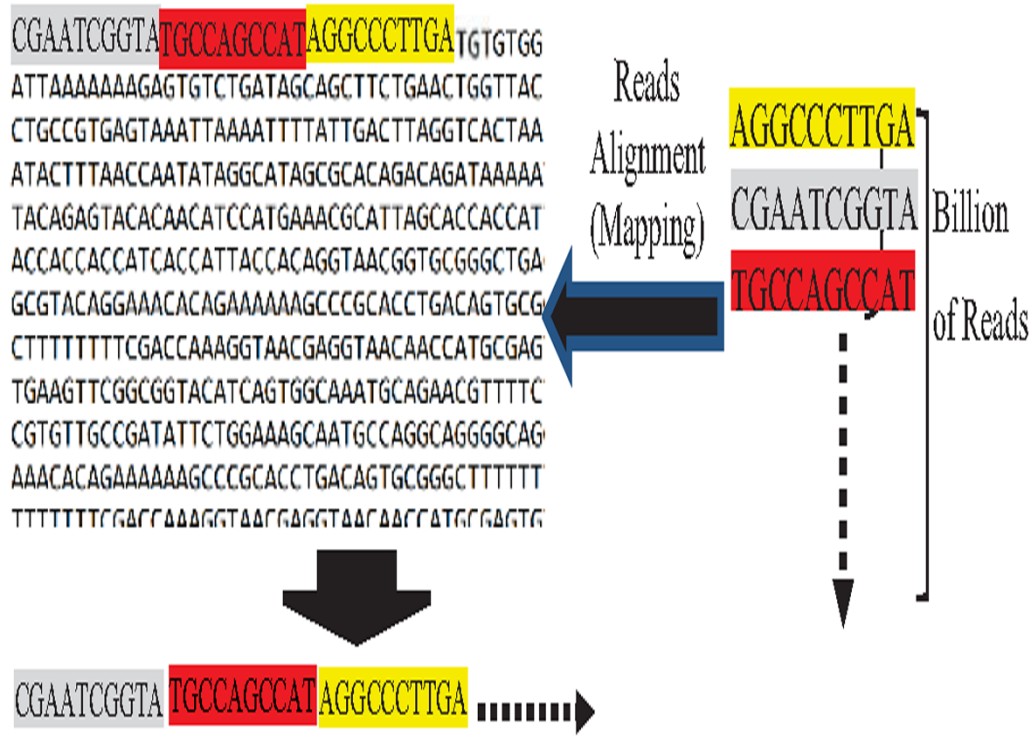

**Figure 5   Reads mapping against reference genome (reads alignment).**

Most *de novo* overlapping algorithms have a high degree of misassembly resulting from repeat ambiguity. The greedy algorithm is one of the simplest algorithms in a data structure. It is used for overlapping. It starts with joining the two reads that overlap the highest. Unassembled reads are chosen for the next run until a pre-defined minimum threshold is reached (*Simpson & Pop, 2015*). Although the greedy approach is computationally feasible, it cannot distinguish the repeats, merging reads wrongly resulting in misassembly (*Simpson & Pop, 2015*) as shown in the example in Fig. 8 for the Genome G = {GGATGGGGATGCCT} that has two repeats ''ATG''. Overlap Layout Consensus (OLC) is a famous graph-based algorithm introduced by Staden in 1980 and subsequently improved and extended by others. Many *de novo* assembly tools are based on OLC methods, such as Newbler, PCAP, Celera Celera, CAP3, and ARACHNE. In OLC, repeats in short reads create a mathematical problem of finding a Hamiltonian path. The graph will be too complex if the genome is highly repetitive, such as the human genome.

Another widely used algorithm is De Bruijn Graphs (DBG), which Nicholas Govert de Bruijn introduced in the 1940s. DBG considers each read is a k-mer from the Genome (*Simpson & Pop, 2015*). The first step in DBG is to divide each read with length k into two substrings (edges) with length (k-1). Next, draw each left substring corresponding to its right substring. A path that contains every edge of the graph exactly once is called the Eulerian path or Eulerian walk. DBG is a widely used algorithm. Current *de novo* assemblers use DBG, such as Euler, Velvet, SOA Pdenovo, ABySS, and IDBA. However,

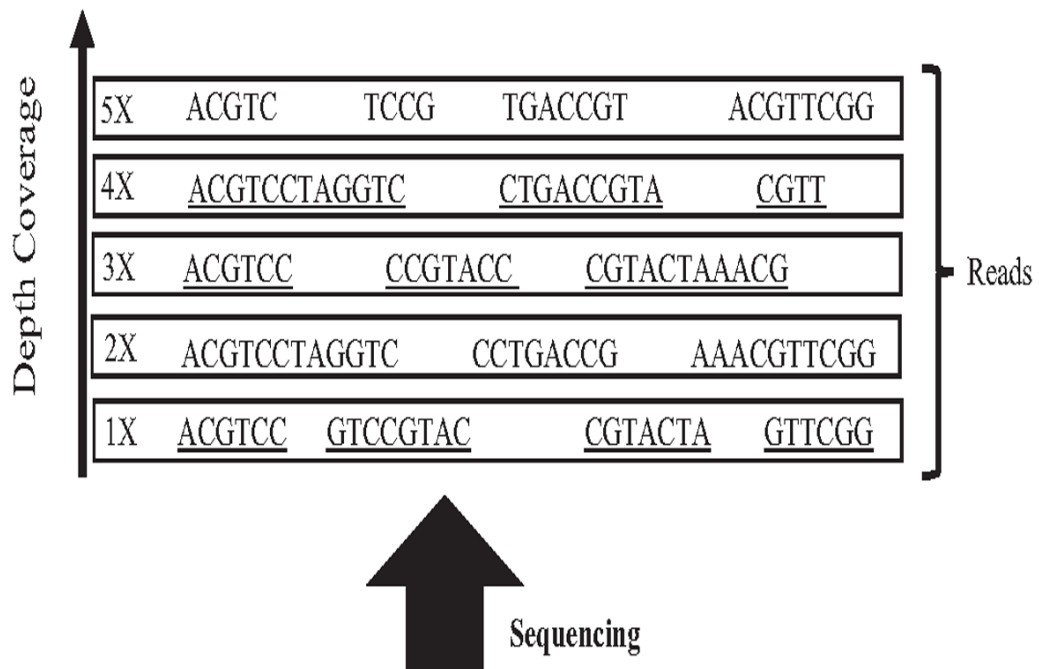

**Figure 6** Sequencing coverage depth.

repeats in short reads create a fragmented assembly problem in this algorithm, shown in Fig. 9.

In the reference-guided approach, repetitive sequences can result in ambiguity during the alignment process when a read has approximately matches with many positions (offsets) in the reference genome. This ambiguity is a well-known data structure problem called the multiple-matching problem. However, multiple-matching results in misassembly, as shown in Fig. 10. Most alignment tools cannot accurately map reads against the reference genome. The sensitivity or confidence of the Alignment tool (S) is calculated by the ratio of correctly mapped read to incorrectly mapped read at a particular threshold (S = Number of reads mapped correctly/a number of reads mapped incorrectly). Repeats in short reads affect the sensitivity of the alignment tool, resulting in a degree of misassembly, decreasing the total accuracy of assembly (*Thankaswamy-Kosalai, Sen & Nookaew, 2017*).

### Computational performance challenges in genome assembly

The second computational problem in genome assembly is challenging performance due to the massive data amount generated by high coverage NGS. This massive data amount generates big performance challenges in each assembly approach in terms of CPU, memory, and storage. Assembly performance challenges exist in both assembly approaches (*Jain et al., 2018b*). In the reference-guided approach, reads alignment works in time complexity O(NL). N is the number of reads (millions) to be aligned against each offset of reference

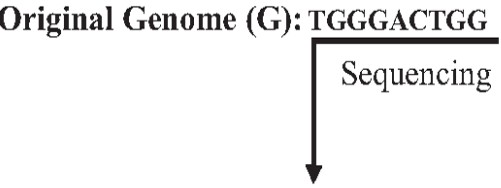

**Original Genome (G): TGGGACTGG**

Sequencing

*Input from biology to computer Science:*

S = {GACT, ACTGG, TGGGAC }

Overlapping algorithm

*Output from computer science to biology:*

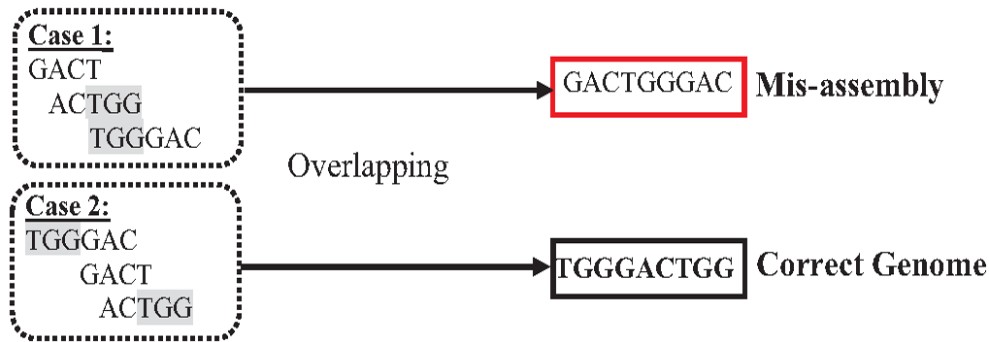

**Case 1:**
GACT
  ACTGG
    TGGGAC

Overlapping

GACTGGGAC **Mis-assembly**

**Case 2:**
TGGGAC
  GACT
    ACTGG

TGGGACTGG **Correct Genome**

**Figure 7**   **Mis-assembly in *de novo* approach.**

genome with length L, which might be billions of letters. It is a highly intensive process, as shown in Fig. 11. On the other hand, computational complexity in *de novo* overlapping works in time complexity $O(N)^2$, where (N) is the number of reads (millions). Comparing millions of reads against millions to detect overlapping is a well-known data structure querying problem called all-against-all or all-suffix-prefix problem (ASPP), as shown in Fig. 11 (*Haj Rachid, 2017*).

### Accuracy and performance evaluation in genome assembly

The accuracy of the genome assembly produced through the assembly approaches can be evaluated using metrics scores throughout the assembly process, such as counting, the number of contigs, the proportion of reads that can be aligned against the known reference genome if exists (misassembly degree), and the absolute length of contigs. A commonly used statistical metric is N50. Given a set of contigs, each with its own length, the N50 is defined as the shortest contigs' length in the group of contigs which represent 50% of the assembly length. N50 can be used to denote the contiguity of the assembly. The pseudocode of how to calculate N50 from an array of contigs is shown in the pseudocode in Box 1 (*Manchanda et al., 2020*). Although N50 is still widely used for assembly evaluation, this metric does

**Figure 8  Mis-assembly in greedy algorithm.**

not reflect the quality of an assembly and can be misleading (*Lischer & Shimizu, 2017*). An example of using N50 to evaluate the genome assembly is a *Ciona intestinalis* genome with an estimated N50 of 234 kb length, which was found inaccurate a few years later due to repeat problems that resulted in misassembly. The N50 metric does not consider that some contigs may be erroneously joined or even overlapped (*Giordano et al., 2018*; *Castro & Ng, 2017*). U50 is similar to N50 in the calculation, and the only difference is that U50 uses unique contigs for calculation that do not overlap with other contigs to represent a more accurate result (*Lischer & Shimizu, 2017*). Another method to ensure the completeness of an assembly is by detecting the presence-absence variations (PAV) against the reference genome. It identifies the sequences in the reference genome that are entirely missing in the newly generated assembly (*Giordano et al., 2018*). Also, QUAST-LG is a tool that compares large genomic *de novo* assemblies against reference sequences and calculates the quality metrics (*Mikheenko et al., 2018*). The Merqury, is a reference-free tool assembly evaluation By comparing k-mers in a *de novo* assembly to those found in unassembled high-accuracy reads (*Rhie et al., 2020*; *Chen et al., 2021*).

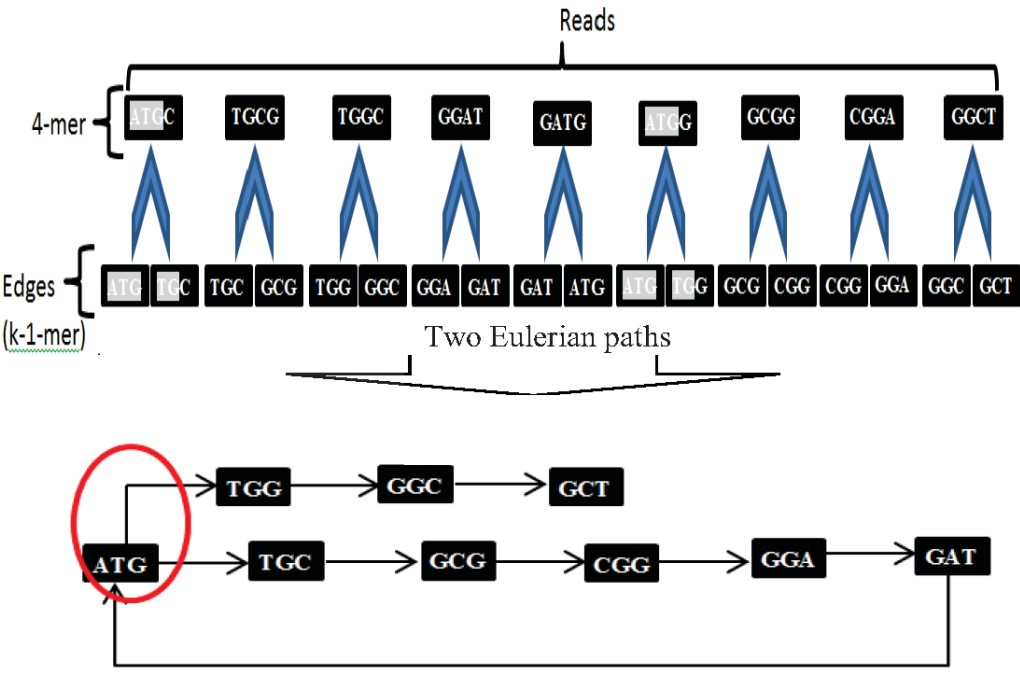

**Figure 9 Fragmented assembly in DBG algorithm.**

**Box 1. N50 Metric Calculation**

Let C array of scaffolds lengths C ={c1, c2, c3,…,ci}
Let AL is Sum C[]
Let AL50 = Sum C[]/2
Let incL is the incremental length with initial value incL =0
Let CN50[] is an empty array to store N50 scaffolds' members
Let i is the length of array C[]
Sort C descending
For n =1 to n =i
Do
IF (incL = AL50)
Break For Loop
ELSE

incL = incL + C[n].Length
Add C[n] to CN50[]
End IF

*Input from biology to computer Science:*

R={GGATGCGATGGCT} Reference Genome--------------Input 1

R={ATGCG, GAGTT, ATGCA } read to be aligned--------Input 2

Input String: ATGCG, GAGTT, ATGCA

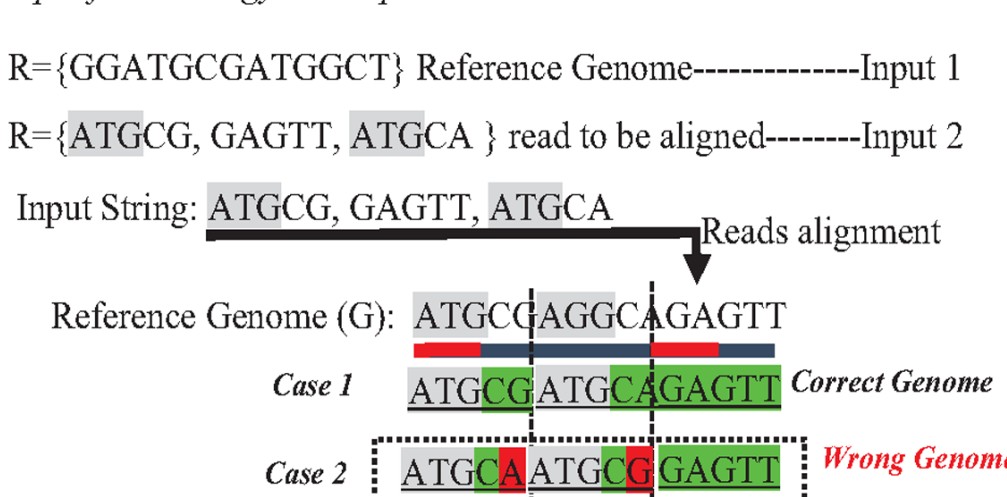

*Output from computer science to biology (if case 2):*

ATGCAATGCGGAGTT which is wrong genome

**Figure 10   Mis-assembly in reference-guided approach.**

End For
return CN50[]
Thus:
N50 =min(CN50[])

The performance of genome assembly can also be evaluated through the computational execution time (time complexity), referred to as time complexity, which is represented as O(N), where (N) is the size of the input and (O) is the execution time for the algorithm, it can be linear, logarithmic, quadratic, exponential or any other relationship. For example, if the number of reads to search for overlapping is five reads, and the index hits ten times, the time complexity of overlapping is O(2N), as shown in the pseudocode in Box 2. Also, the computational performance of the genome assembly can also be assessed by memory or storage consumption.

**Box 2.**   Index hit represent the overlapping computational time complexity

Let N =5 reads
Let All-against-all $O(N)^2$ =25 times
Current index hit =10

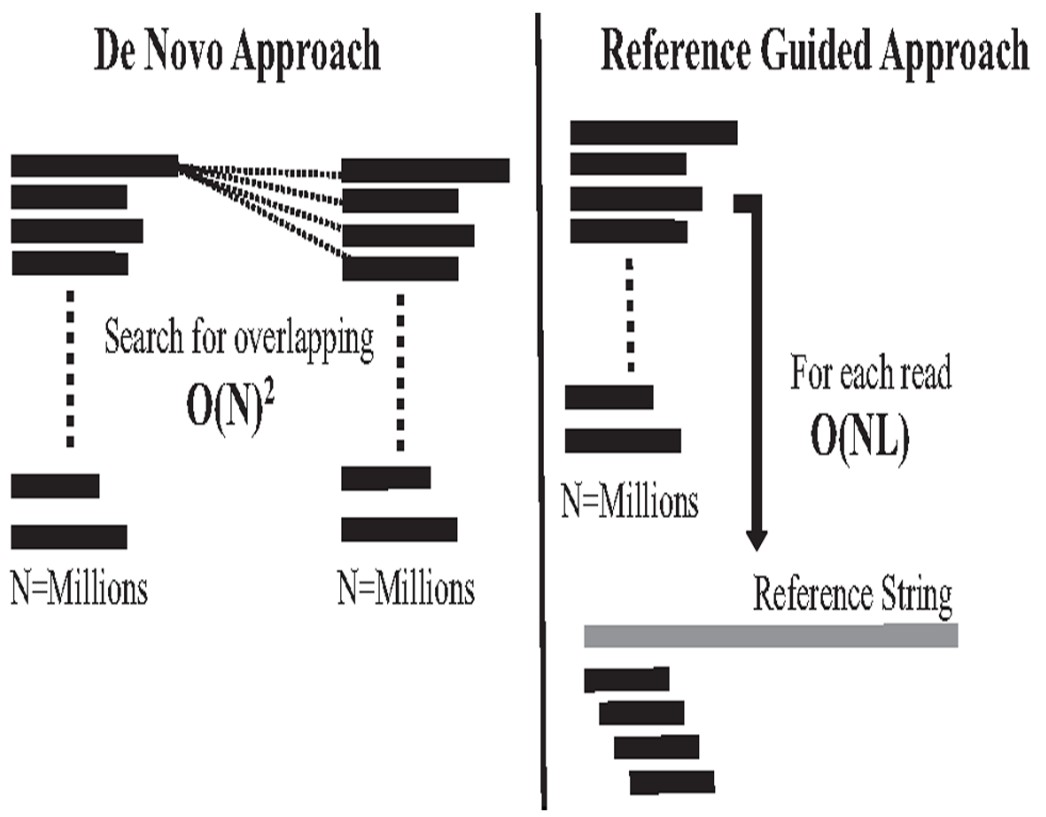

**Figure 11** **Overlapping and reads alignment time complexity.**

Thus: Overlapping thus computational time complexity is O(2N) which is liner algorithm.

## Survey methodology

This review article summarizes the enhancement in genome assembly accuracy using prior repeat identification and hybrid assembly approaches. Also, we highlight the limitations of current methods. On the other hand, we overview the most famous data structure indexing to optimize genome assembly, such as k-mer index, tree index, hash index, also parallelization methods. We also highlight the limitations and show that combining more than one method improves accuracy and performance. This review helps computer science researchers understand genome data and its considerations, such as repeat. Also, it will help to identify the main research problems in the genome assembly area in terms of accuracy and performance optimization.

### Search strategy

The authors found the primary studies through seven digital libraries: Web of Science, Science Direct, SCOPUS, Google Scholar, IEEE Explore, PubMed, and EMBASE. These digital libraries are most preferred and widely used by the research community. The search

query string is applied to an individual database to find relevant studies based on the research questions shown below in Box 3. The search strategy for the literature process is shown in Fig. 12. A total of 65 research papers, after the duplications were removed, were found and downloaded from the mentioned database.

---

**Box 3.** Search string

(Genome assembly OR genome analysis OR indexing genome data OR genome mis-assembly OR parallelize genome data OR repetitive genome sequence identification) AND (accuracy OR performance OR enhancement) AND (method OR approach) AND (*de novo* approach OR reference guided approach OR hybrid approach) AND language (English)

---

***Inclusion criteria***
The studies concerning that are:
1. Written in English
2. Based on genome assembly.
3. Focused on accuracy and performance optimization.
4. Published in conference or journal

***Current Solutions for Misassembly Problem***
**A. Prior Repeat Identification Methods**
Repeat identification is a method to detect and identify repetitive sequences in the reads or in the reference genome (*Zeng et al., 2018*). Identifying repetitive sequences before genome assembly helps the assembly algorithm distinguish the reads that have repeats, and avoid misassembly, thus, significantly enhancing the accuracy of genome assembly from short reads (*Seitz, Hanssen & Nieselt, 2018*). Repeat identification methods are introduced first in the reference-guided approach through the repeat masker method. However, it did not work well with big genomes (*Zeng et al., 2018*). The *de novo* approach is preferable, identifying repeats from target genome reads rather than a reference genome (*Baichoo & Ouzounis, 2017*). Most of the current repeat identification methods are statistically based, calculating the occurrences of repetitive sequences in reads. In order to clearly understand how it works, let (G) is the genome that is sequenced with 5X coverage, so (C =5). Let rs is a repetitive sequence with a length of 9 letters, rs ={ACGTGATAT}. Let R denote the array of reads generated by sequencing genome (G), R ={r1,r2...ri} as shown in Fig. 13. Reads are just a substring of Genome (G). According to the current sequencing coverage, we expect any unique substring of (G) to appear in (R) several times less than or equal to (C). In this example, a read, ri{GTG} appears in the data set six times, which means the frequency of ri, $f$ ri $= 6$, while the sequencing coverage C $= 5$. If ri sequence is a unique sequence in Genome (G) the max $f$ ri $= 5$ under coverage (C $= 5$) and max $f$ ri $= 6$ under coverage (C $= 6$), and so on. If ($f$ rs $= 1$) in (G), it is impossible that the frequency of the entire rs or even any substring of rs exceeds the sequencing coverage (C) in the reads dataset. In Fig. 13, $f$ ri $= 6$, greater than C, simply means that the extra occurrence

of ri comes from sequencing another copy of rs. If the read frequency is less than or equal to the sequencing coverage, it means that this read comes from a unique sequence in the genome. On the other hand, if the frequency of the read exceeds the well-known sequencing coverage, it means that this read comes from the repetitive sequence in the genome. More than half of some genomes are repeats and vary in length, which might be too long, such as TE. Thus, the size of repetitive sequences is used as a criterion to evaluate the accuracy of identifying repeats in many repeat identification methods (*Genovese et al., 2018*; *Taylor et al., 2022*; *Liao et al., 2021*). Many algorithms have been proposed according to the previous statistical concept. Earlier, the RePS algorithm assumed that any 20-mer (substring from the read with length 20 letters) that appears in the dataset more often than a sequencing coverage is likely to be an exact repeat and is masked out. A sliding window assembly (SWA) is a genome assembly algorithm proposed by *Lian et al. (2014)* for identifying and assembling repeats. Fundamentally, the algorithm considers the entire read is a substring of the genome. SWA calculates the frequency of each read in the dataset. Based on the well-known sequencing coverage SWA splits reads into two groups, repeat and non-repeat, assembling them separately. The non-repeat group consists of those reads with a frequency less than the sequencing coverage, while the repeat group includes those with a frequency greater than the sequencing coverage. Then, the overlap is run for each group separately in parallel to construct the genome superstring. However, SWA considers the entire reads as the substring of genome superstring, which means it can detect repeat length that is only greater than or equal to the read length, while repetitive sequences might be smaller than the read's length, such as microsatellite tandem repeat, which cannot be detected by SWA (*Zeng et al., 2018*) as shown in Fig. 14. Repeat *de novo* (REPdenovo) is a repeat identification method proposed by *Chu, Nielsen & Wu (2016)*, similar to the earlier method RepARK that splits reads into smaller substrings (k-mers) then identifies the frequency of k-mers, group them to repeat and non-repeat groups, finally, assembles these k-mers. Unlike SWA, REPdenovo uses the average k-mers frequency as a threshold of this algorithm, where a repeat is identified when a k-mer frequency is more than the threshold, as shown in Fig. 15. REPdenovo is also able to identify a repeat shorter than the read length by constructing them from reads' k-mers to longer repeats. Evaluation and comparison show that REPdenovo outperforms the earlier RepARK method regarding the accuracy and completeness of repeats construction. REPdenovo discovered that previous repeat annotations had missed a significant number of 190 potentially new repeats in the human genome. The high accuracy of identifying the repeat enhances the genome's contiguity with N50 3141, while RepARK method only N50 116. However, when reads are chopped up into k-mers the biological info represented in the reads is lost (*Angeleska, Kleessen & Nikoloski, 2014*). Similar to REPdenovo, the detection of long repeats (DLR) converts all reads into unique k-mers of a certain length and screens out the k-mers with a high frequency (*Liao et al., 2019*). The detection method in the DACCOR assembly is also similar to REPdenovo calculation introduced as a stage in Characterization Reconstruction (DACCOR) hybrid assembly. Unlike REPdenovo, it splits the reference genome into k-mers instead of the reads themselves, then identifies repeats from k-mers of the reference genome, similarly to REPdenovo calculation. The detection method in DACCOR assembly identifies repeats

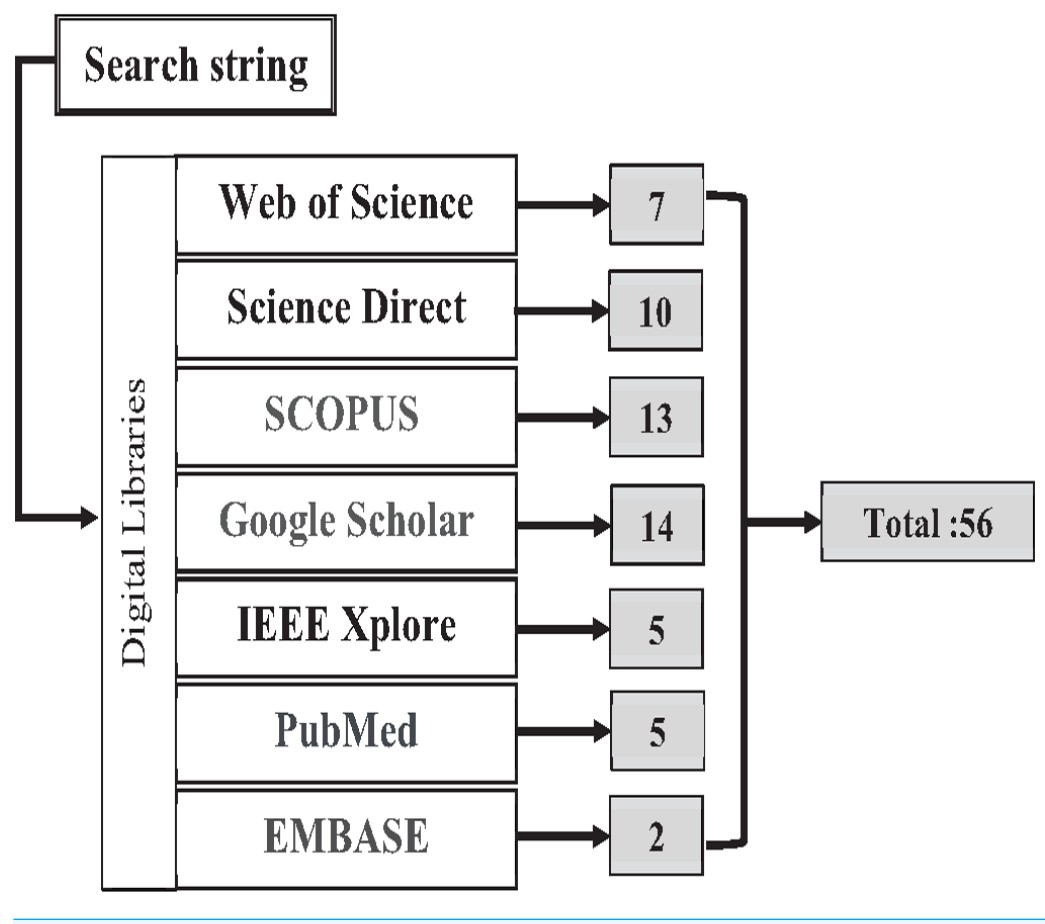

**Figure 12** **Search Map.**

from the reference genome, which might come with low identification accuracy, simply because repeats in the reference genome might be slightly different from the real repeat in target genome reads (*Zeng et al., 2018*).

On the other hand, digital signal processing can also detect repetitive elements. In most signal-processing methods, DNA sequences are converted to numerical sequences, and repetitive elements can be identified by Fourier power (*Yin, 2017*). However, capturing the essential features of repetitive elements, such as copy numbers of repeats, is still challenging. In addition, Fourier transform cannot capture repeats in short genomes (*Yin, 2017*).

Recently, machine learning has been employed for identifying repetitive elements. Repeat Detector (Red) has been proposed by *Girgis, (2015)* as *de novo* tool for discovering repetitive elements in genome reads. Red utilizes a Hidden Markov model (HMM) dependent on labeled training data, and it successfully identified new repeats in the human genome. However, it only generates genome coordinates for repeats without any annotation. Therefore, the red output does not help analyze repeat content or TEs evolution (*Zeng et al., 2018*). Improvements are required in optimizing computation requirements and choosing suitable training datasets for such a machine-learning program

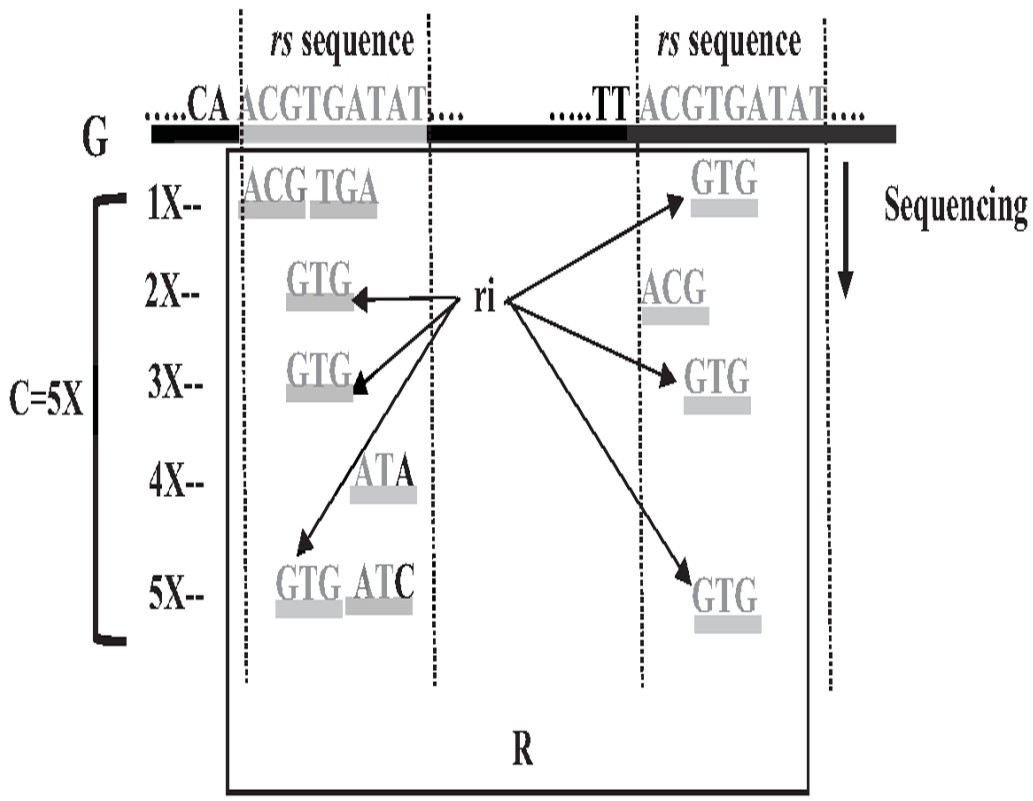

**Figure 13 Repetitive sequences frequency greater than coverage depth.**

in order to strengthen the program accuracy and precision on the repeat prediction in the genomes (*Libbrecht & Noble, 2015*).

As shown in the current limitations of some current repeat identification methods in Table 1. No one algorithm fits all lengths of repetitive sequences (*Guo et al., 2018*). The combination of two or more methods might introduce a better solution to identify and assemble repeat accurately. A combination of repeat identification methods consistently outperforms the usage of a single method due to the variety of repetitive sequences. An example of this combination is merging consensus sequences generated by RepeatModeler and Repbase library, which successfully annotated many repeats in many genomes (*Zeng et al., 2018*). Another example of a repeat identification methods combination is Tandem Repeats Finder (TRFi), used to identify tandem repeats and combined with the RepeatExplorer method, and the result outperforms both of them in identifying complex tandem repetitive sequences (*Peška et al., 2017*). However, combining methods adds more computational complexity, which might be addressed by data indexing and increasing the parallelism level during the identification process.

**B. Hybrid assembly approach**

The last two decades have seen significant improvement in solving the misassembly problem that resulted from repeats through the hybridization concept. Fundamentally,

**Figure 14** Repetitive sequence as substring of read.

hybridization combines two or more methods to perform genome assembly. Hybridization appears in genome assembly in different ways. It can combine data from different platforms (*Chen, DL & Meng, 2020*) or two or more genome assembly methods. The example is combining the OLC and the DBG method to introduce DBG2OLC to enhance assembly accuracy that is affected by repeats (*Jain et al., 2018a*). The hybridization between *de novo* and reference-guided approaches creates a hybrid assembly hybrid approach which able to complement each approach's limitation (*Platt, Blanco-Berdugo & Ray, 2016*). The idea of a hybrid assembly approach has been introduced by *Silva et al. (2013),* who proposed a scaffold-builder assembler composing an initial *de novo* assembly from reads, followed by aligning contigs against a reference genome to build a scaffold. Enhancing assembly accuracy through the hybrid assembly approach adds many contributions to understand the diversity of species. An example of is a study on Cyclospora cayetanensis species, a parasite that caused intestinal infection, has a highly repetitive genome that could not be assembled accurately using *de novo* alone. Combining a reference-guided approach with a *de novo* approach generates new assembly for this species with a significantly greater depth of coverage and a lower degree of misassembly (*Gopinath et al., 2018*).

Although, accurate reconstruction of repetitive sequences cannot be without repeat resolution (repeat identification). Most of the current assembly hybrid approaches do not employ prior repeat identification (*Seitz, Hanssen & Nieselt, 2018*). The accuracy is enhanced by using more than one reference genomes or multiple samples of the target genome from different sequencing platforms (*Gopinath et al., 2018*). The characterization and reconstruction of repetitive regions (DACCOR) introduced a new idea to enhance

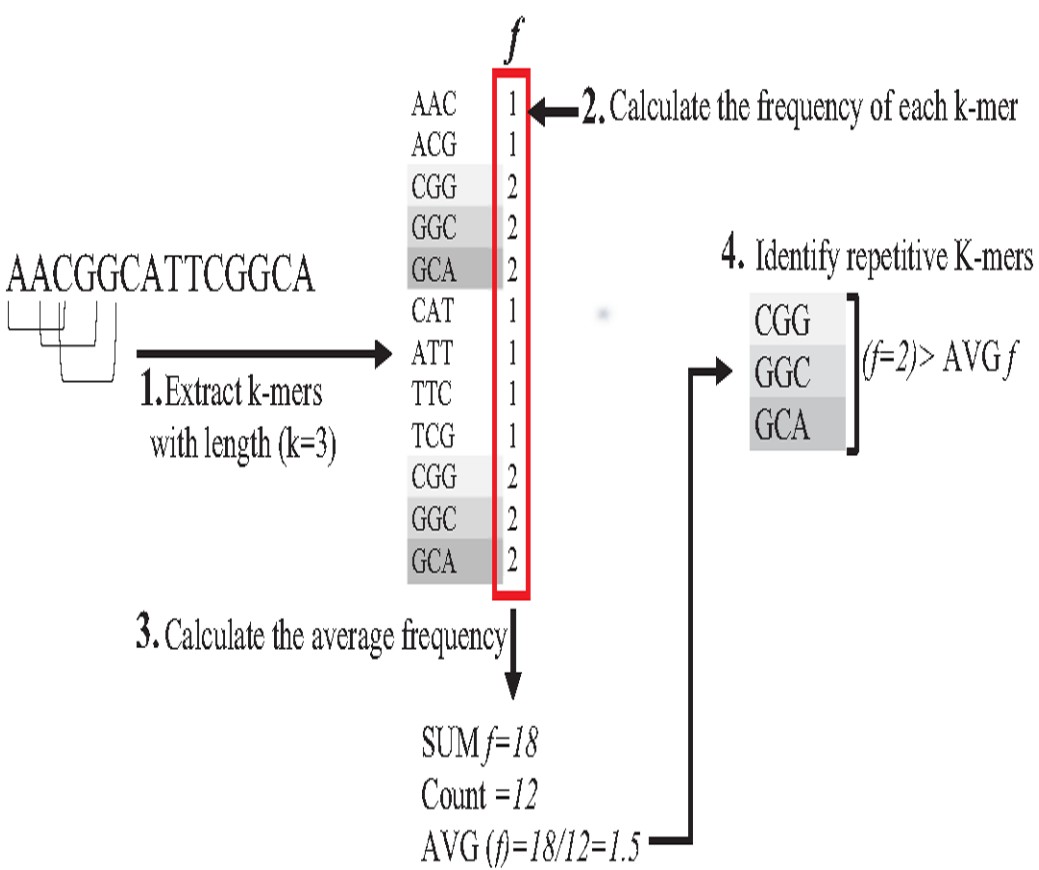

**Figure 15** Reads',figK-mers Frequency in REPdenovo.

**Table 1** The limitations of current repeat identification methods.

| Repeat identification method | Limitations |
| --- | --- |
| SWA | Cannot detect repetitive sequence shorter than read's length |
| Detection in DACCOR assembly | Identify repetitive sequences from the reference genome that are slightly different from the target genome resulting in low identification accuracy. |
| REPdenovo | Biological info represented in read structure are lost when the reads are chopped up into k-mers |
| Digital signal processing | Difficult to capture a copy number of repeats and cannot capture repeats in short DNA sequences which means that many types of tandem repeats cannot be identified |
| Machine learning | Not useful for analysing repeat content such as transposon element for biological evolution and big performance challenge on massive genome data |

the hybrid assembly approach based on identifying repetitive sequences in the reference genome prior to hybrid assembly. Although the detection repeat identification method in DACCOR identifies repetitive sequences from the reference genome, the results on

the bacterial genome *Treponema pallidum,* as shown in Table 2, reveals high accuracy in identifying repetitive sequences (*Seitz, Hanssen & Nieselt, 2018*). In Table 3, the comparison of the accuracy in generating contigs from repetitive regions detecting repeats for T. pallidum (sample AR1 with the coverage X157), DACCOR successfully constructed 96.8% of contigs that can be mapped to the reference genome, while, SPAdes *de novo* assembly generates about 69.5% of contigs from repetitive regions that can be mapped to the reference genome and EAGER reference guided approach with generates around 42.1%. Using repetitive sequences from the reference genome to detect repeats in the target genome in DACCOR might have advantages over the conventional assembly approach. However, the repeats identified through the approach might be limited to the reference genome, which may not well represent the actual repetitive sequences present in the target genome. On the other hand, the hybrid assembly approach comes with big performance challenges. The total assembly time in the hybrid approach is exceptionally challenging for any traditional algorithm or hardware architecture. Although performance challenges are well-optimized in each approach individually (*Liu et al., 2018*), no data structure indexing is introduced to fit overlapping and reads alignment together in the hybrid assembly approach.

## Current solution for performance optimization of genome data
### A. Indexing in genome assembly

Indexing could speed up the search by reducing the number of iterations (time complexity), leading the algorithm to focus on a specific part of the data instead of searching in the entire dataset (*Xiaolei et al., 2015*). One of the earliest data structure indexing methods is k-mer index. However, massive genome data requires more optimization for this index. The keys of k-mer index can be extracted as a shape, skipping letter instead of extracting all k-mers from the string to speed up the index hitting. As shown in Fig. 16, k-mers are extracted as a pattern by taking the first and third letters followed by the fifth and sixth letters. This reduces the index total size, which improves the specificity of the index hitting and index verification. This kind of index must be hit with the same index pattern. This idea was implemented in Homopolymer Compressed k-mers method proposed by (*Liu et al., 2018*). This implementation added significant enhancement of mapping reads to the reference genome. Another variation to speed up querying k-mer index is a binary search. Binary search, also known as half-interval search, logarithmic search is a search algorithm that finds the position of a target value within a sorted array. The k-mers of the genome must be ordered alphabetically in ascending order, as shown in Fig. 17. In this case, the query does not look up in the entire index simply because (TGG) is alphabetically greater than (GTG). The query does bisection (dividing the problem into two halves), skipping the first part of the index (*Berztiss, 2014*). Each iteration bisection occurs, resulting in a significant reduce the query time. The total number of bisections that are needed to perform can be calculated as $Log_2(n)$. This implementation significantly reduces search trials during reads

**Table 2    Repeat identification in *Treponema pallidum* Genome using Detection method in DACCOR.**

| Detected repetitive sequences | *Treponema pallidum* |
|---|---|
| True positives | 22,382 |
| True negatives | 1,116,478 |
| False positives | 0 |
| False negatives | 773 |
| Accuracy(%) | 99.93 |

**Table 3    DACCOR Enhance the genome resolution compared to SPAdes and EAGER method.**

| Sample | Method | Contigs from repetitive regions mapped against reference genome |
|---|---|---|
| ARI (Coverage 157X) | SPAdes (*de novo*) | 69.5% |
| | EAGER (Reference-guided approach) | 42.1% |
| | DACCOR (Hybrid approach with prior repeat Identification | 96.8% |

mapping to reference genome and can be implemented for reads during the search for overlapping (*Brodsky et al., 2010*).

The Hash index is the most famous implementation of k-mer index. In the hash index, the hash function is responsible for mapping each distinct k-mer to one bucket (group), as shown in Fig. 18. the Hash index is widely used in many studies on genome data for indexing dataset reads (*Zhang et al., 2018*) to reduce suffix-prefix search in *de novo* overlapping. However, because of repeat, sometimes two keys may generate an identical hash causing both keys to point to the same bucket, which is known as a hash collision, slowing down the search in the hash index (*Xiaolei et al., 2015*).

Suffix indexes are another data structure index family. There are four variations of suffix indexes: trie suffix tree, suffix array, and FM index. However, the suffix tree (ST) is the most popular and widely used indexing tool in bioinformatics applications (*Barsky et al., 2009*). It is the base of some popular sequence alignment tools, such MUMmer and RePuter, and can be used to solve more complex problems such as repeat identification. Unfortunately, the construction of ST is highly memory and CPU-consuming (*Pingali, Tanay & Baruah, 2017*). However, there are many research efforts to reduce the construction of ST time complexity, for example, scaling the construction of ST on multiple CPU cores (64 cores) that (*Labeit, Shun & Blelloch, 2017*) achieves a speedup from 2X to 4X over the original algorithm to construct S.T. Figure 19 shows the construction of ST index from genome G ={ACGCGT}.

In overlapping massive genome reads, maybe performing an approximate match between suffix and prefix first add a significant reduction of time complexity of overlapping by skipping the exact match for suffix prefix candidates that do not have an approximate match. The pigeonhole principle (PP) is a well-known data structure principle called seed and extends principle that can achieve this idea. PP separates the read into non-overlapping partitions, and the exact matching algorithm checks the match of one of the partitions

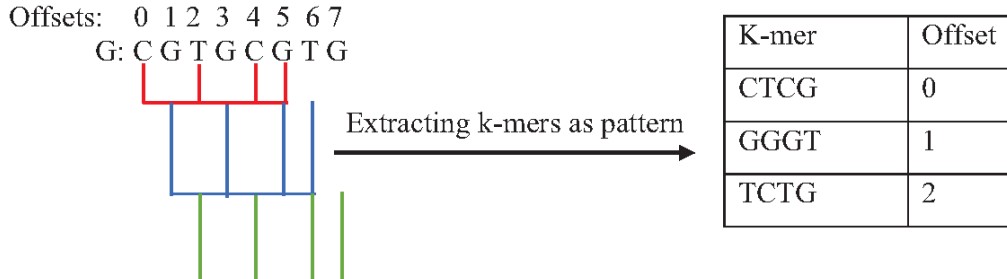

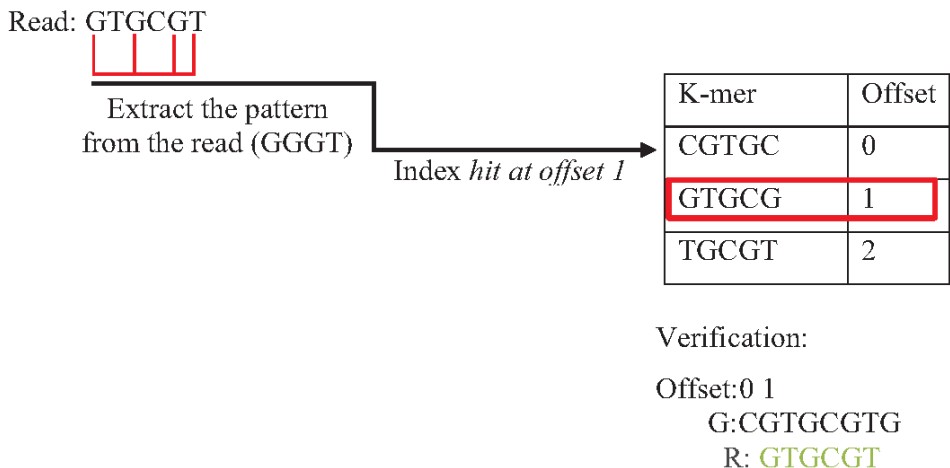

**Figure 16** Querying k-mer index with pattern match.

considered suffix prefix approximate match, then the verification step must be done as shown in Fig. 20. According to this powerful principle, many algorithms and methods for indexing introduced to genome data took advantage of the pigeonhole principle and suffix-tree index. The result of *Escherichia Coli* genome shows that the pigeonhole solution with prefix tree is superior in terms of time and storage compared to the traditional suffix tree for overlapping (*Haj Rachid, 2017*), as shown in Table 4.

The optimization using data structure indexes for overlapping reads and read alignment in the genome hybrid assembly approaches remains a challenge. However, ST is the best solution for repetitive sequences scenarios, its construction memory, and CPU consumption. In Hash, index collisions slow down the search at this key. FM.- index is based on Burrows-Wheeler Transformation (BWT) that was proposed by Burrows and Wheeler in 1994 (*Wang et al., 2002*). Querying FM index cannot use binary search because it does not consist of the complete rotation of the genome and needs another process to find the offset of the index hit, which is a highly computational task in massive genome data.

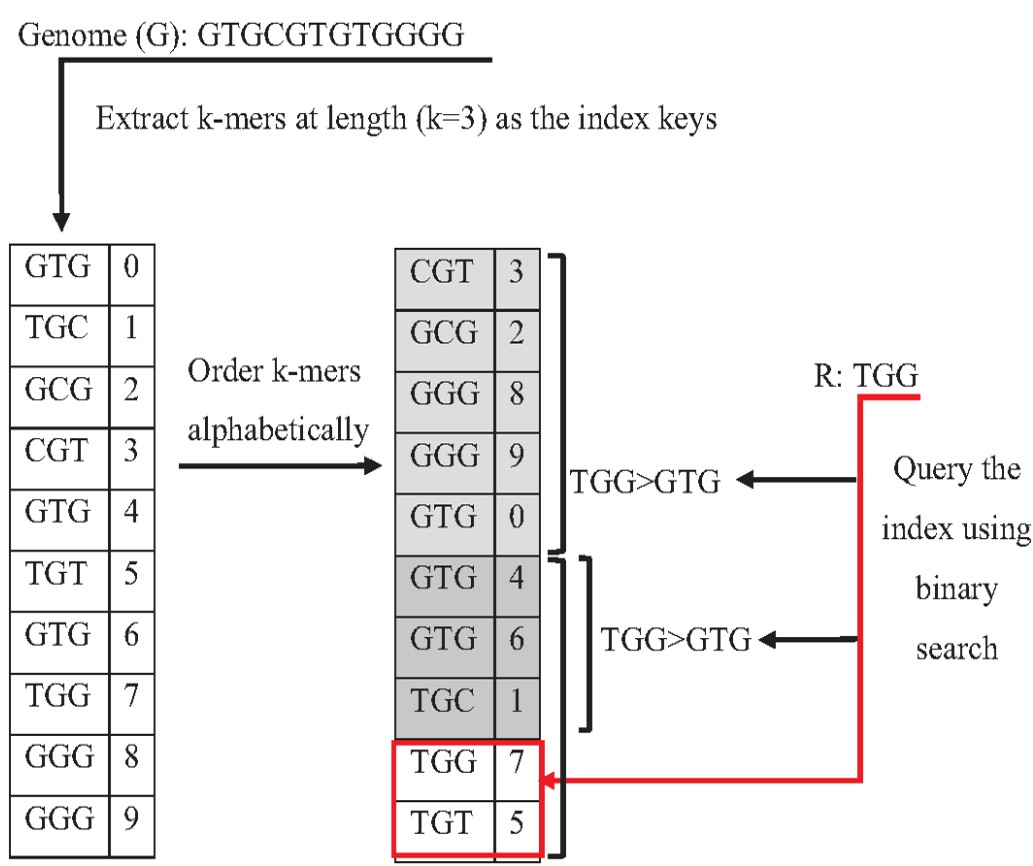

**Figure 17** Binary search to querying k-mer index.

Hybridization between two indexes might introduce a better solution for indexing massive genome data. An example of the hybrid between two indexes is MiniSR index which uses k-mer indexing with the hash index. MiniSR can index the massive human genome in a few minutes (*Bayat et al., 2018*). The hybridization between a few indexed might introduce a solution to optimize overlapping and read alignment in hybrid genome assembly.

## B. Parallelization in genome assembly

The exponential growth of genomic data has instigated a significant challenge for genomics analysis computing infrastructure and software algorithms. Genomic experiments are now reaching the size of Terabytes and Petabytes (*Shi & Wang, 2019*). Scientists may require weeks or months to process this massive amount of data using their own workstations (*Shi & Wang, 2019*). Parallelism techniques and high-performance computing (HPC) environments can help to reduce the total processing time (*Ocaña & De Oliveira, 2015*; *Shi & Wang, 2019*). The simplest definition of parallelization is to split the extensive process into smaller subprocesses and run them in parallel on multiple CPUs (*Pingali, Tanay & Baruah, 2017*; *Shi & Wang, 2019*). It solved many performance

**Figure 18** Construction of hash index and collision resulted by repeat.

challenges in the big data area. However, parallelization is introduced in genome data for genome assembly in some stages individually.

Sequencing technologies produce millions to billions of short reads. The first step to assembling these reads is to extract them from FASTA or FASTQ files into structured database tables, which is too time-consuming. Parallelization could reduce raw data streaming time is to split big FASTA or FASTAQ files into smaller files processed in parallel (*Pan et al., 2016*).

Another variation of parallelization in genome data is parallelizing the index's construction. An example of this idea is proposed by *Haj Rachid & Malluhi (2015)* for speeding up prefix tree index construction by letting each processor work on strings that start with a specific character in the alphabet (A, C, G, and T). For example, processor 1 constructs the part of the tree that corresponds to strings that start with the letter "A" while processors 2, 3, and 4 construct the parts of the tree corresponding to the strings starting with "C", "G", and "T" respectively. A similar idea was introduced by (*Ellis et al., 2017*), proposing HipMer assembler to construct k-mer index using a deterministic function to map each k-mer to a target processor.

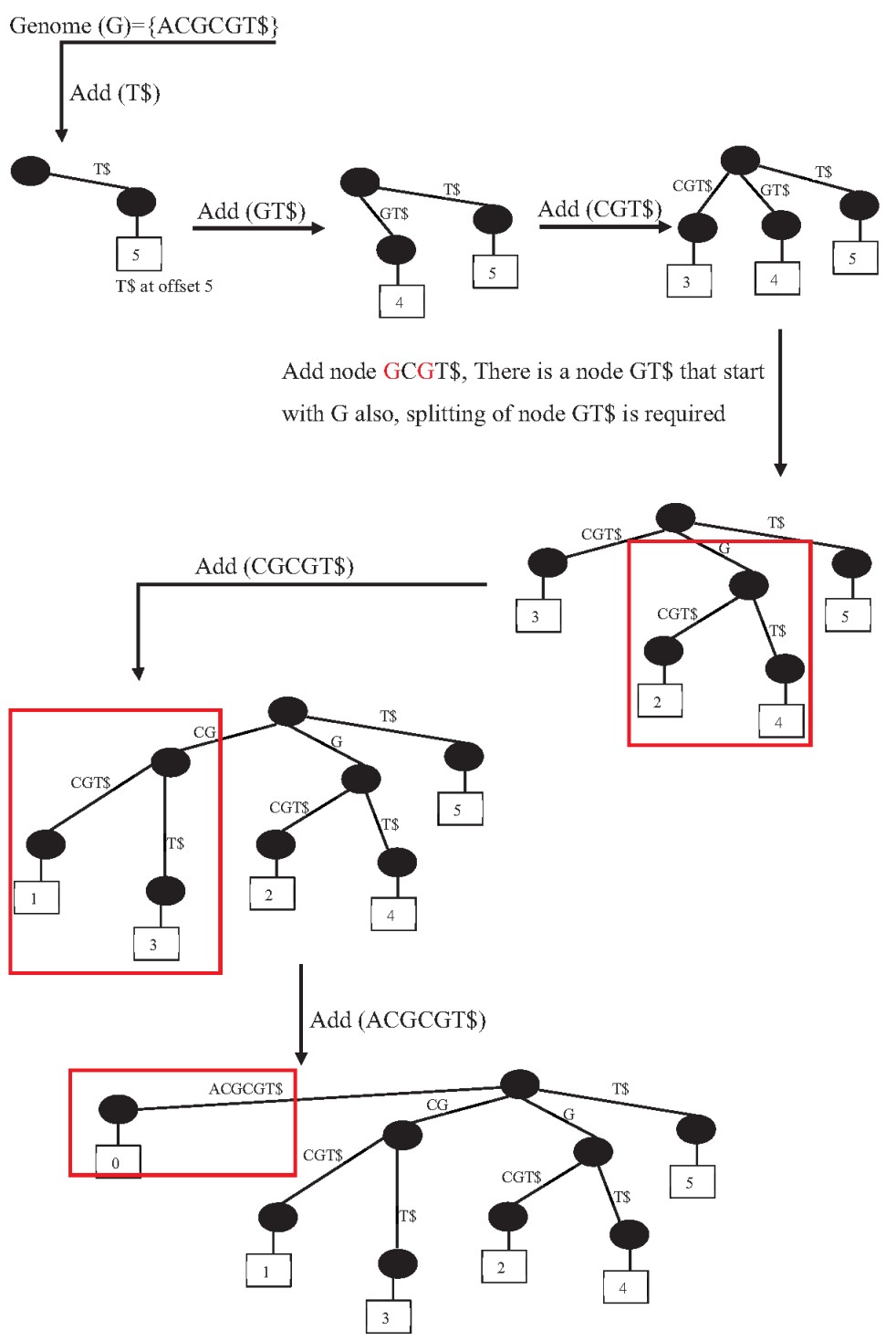

**Figure 19  Prefix-tree index construction.**

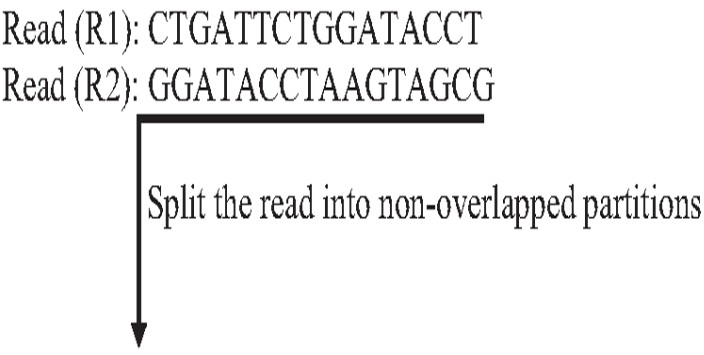

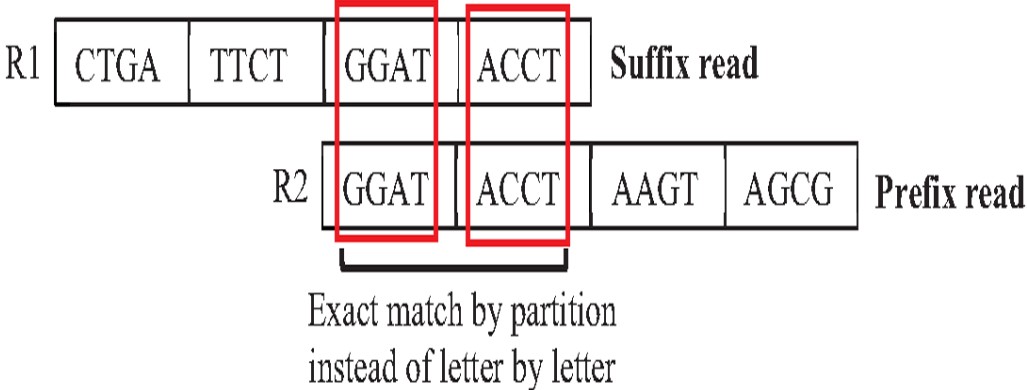

**Figure 20  Pigeonhole principle in overlapping.**

**Table 4  Comparison betdeegPT, PP index in time and space.**

| Metric | PT | PP |
|---|---|---|
| Space consumption | 230 MB | 298 MB |
| Time consumption | 757 second | 49 second |

In de nove overlapping and read alignment of reference-guided approach, the IBD algorithm addressed pairwise comparison problems that can be used to overlap two reads or align reads against k-mer of the reference genome (*Sapin & Keller, 2021*). The algorithm breaks the problem ($N^2$) using massive parallelization with each order (N) comparison. This approach is applied to the U.K. Biobank dataset, with a 250X faster time and 750X less memory usage over the standard approach of pairwise alignment.

Over the last decades, many computing platforms parallelize different stages of genome assembly. For instance, Hadoop (*Shi & Wang, 2019*) introduced a powerful idea of parallelizing the suffix tree index construction. However, the Apache spark platform works 100 times faster than Hadoop, especially in iterative operators (*Wan & Zou, 2017*). A distributed and parallel computing tool named HAlign-II was introduced by *Shi & Wang (2019)* to address read alignment with extremely high memory efficiency.

GraphSeq method proposed splitting genome files and using a high scaling platform (*Su et al., 2018*). GraphSeq works on the Apache spark platform and achieved 13X speedup *de novo* genome assembly. GraphSeq splits a big compressed file into several small ones loading all of the reads in parallel, generating the corresponding suffixes, grouping those suffixes with the exact initial string into the same partition, and applying string graph construction in parallel by partitions. Similarly, an innovative algorithmic approach proposed by *Paulson (2018)* is called Scalable Overlap-graph Reduction Algorithms (SORA), performing string graph reduction on Apache Spark. SORA was evaluated with human genome samples to process a nearly one billion edge of string graph in a short time frame with linear scaling.

Parallelization enhances the performance of streaming massive genome data, genome index construction, and overlapping in the *de novo* approach or read alignment in the reference-guided approach. Despite its performance, due to the nature of the hybrid assembly approach, to our best knowledge, the usage of parallelization in overlapping and reads alignment for genome assembly is yet to be fully implemented.

# CONCLUSIONS

Assembling genome reads with the presence of repetitive sequences at a good quality is essentially important but, indeed, far more challenging. Ignoring repeats in the genome would revoke the purpose of the genome assembly. A massive number of reads produced by sequencing technologies have caused high computational performance for the genome assembly approach. A lot of research has contributed significantly to the development of enhancing assembly accuracy by reducing the degree of misassembly in the genome assembly approaches. In this article, we suggest by using prior repeat identification methods to create prior repeat resolution guiding overlapping algorithm in *de novo* or alignment algorithm in the reference-guided approach could reduce the chance of genome misassembly. However, none of the repeat identification methods can accurately detect all types with different lengths of repetitive sequences. The combination of *de novo* with the reference-guided assembly approach could yield better results for genome assembly, which can be another alternative solution. Although the hybrid assembly approaches found to outperform the individual *de novo* approach and the reference-guided approach, repetitive sequences can still generate misassembly, which could be enhanced through the use of multiple samples or multiple reference genomes. However, using multiple samples or reference genomes will add high computational complexity to the hybrid assembly approach. Moreover, most of the current assembly hybrid approaches still lack repeat identification prior to the approach, which can help to increase the accuracy of the genome assembled.

The combination between the *de novo* approach and the reference-guide approach comes with a more computationally extensive performance challenge than using the approach individually. Therefore, the investigation of optimizing the computational performance challenges in the hybrid assembly approach for overlapping and reads alignment is still an open challenge.

Enhancement of the repeat identification accuracy by approaches would be one of the main priorities in addressing issues of genome misassembly. The enhanced repeat

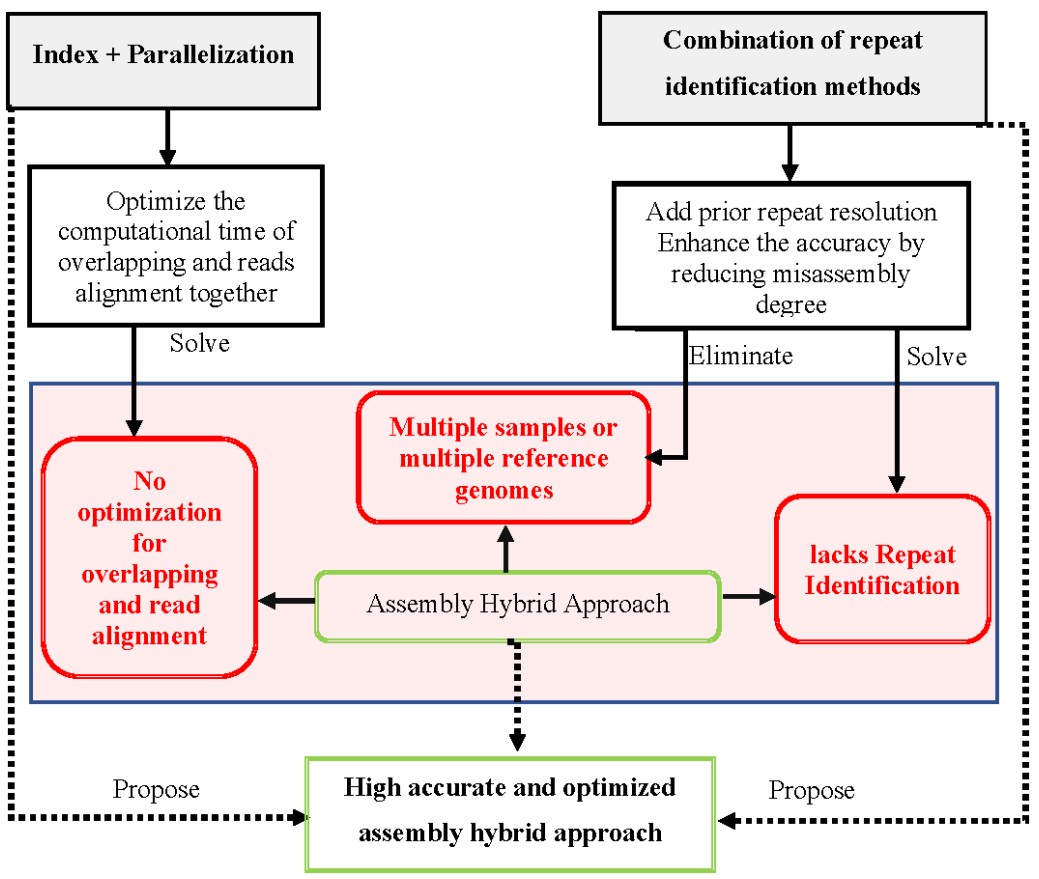

**Figure 21** Solution to propose accurate and optimized assembly hybrid approach.

identification method should be added at the prior stage for hybrid genome assembly to enhance the accuracy without relying on multiple samples or reference genomes, which can increase the computational complexity of the the hybrid assembly approach.

Consequently, the performance of the hybrid assembly approach might be optimized through the hybridization of indexing methods with parallelization to optimize overlapping and reads alignment in the hybrid genome assembly approach.

Hybrid genome assembly enhanced by accurate prior repeat identification and optimized by a combination of indexing and parallelization, as shown in Fig. 21, might be a novel solution for assembling repetitive genomes from short reads.

### Funding
This research was supported by the Ministry of Higher Education (MOHE) through Fundamental Research Grant Scheme RGS/1/2019/ICT02/UTM/02/13, Research Management Centre (RMC), UTM, and ALI@S research group. The funders had no

role in study design, data collection and analysis, decision to publish, or preparation of the manuscript.

## Grant Disclosures

The following grant information was disclosed by the authors:

The Ministry of Higher Education (MOHE) through Fundamental Research Grant Scheme: RGS/1/2019/ICT02/UTM/02/13.

Research Management Centre (RMC), UTM, and ALI@S research group.

## Competing Interests

The authors declare there are no competing interests.

## Author Contributions

- Sherif Magdy Mohamed Abdelaziz Barakat conceived and designed the experiments, performed the experiments, analyzed the data, prepared figures and/or tables, and approved the final draft.
- Roselina Sallehuddin performed the experiments, authored or reviewed drafts of the article, and approved the final draft.
- Siti Sophiayati Yuhaniz performed the experiments, prepared figures and/or tables, authored or reviewed drafts of the article, and approved the final draft.
- Raja Farhana R. Khairuddin performed the experiments, prepared figures and/or tables, authored or reviewed drafts of the article, and approved the final draft.
- Yasir Mahmood analyzed the data, authored or reviewed drafts of the article, review the language and the context of the article, and approved the final draft.

## Data Availability

This is a literature review.

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
