# Peer review of "Genome assembly composition of the String “ACGT” array: a review of data structure accuracy and performance challenges"

_PeerJ Computer Science, doi:10.7717/peerj-cs.1180_

## Round 0.1 · original submission · Minor Revisions

Please, consider seriously the suggestions of both reviewers and refine your manuscript. You may ignore those requirements of the reviewers that demand too much work, i.e. Ref#2's first point under 'Validity of Findings'

Reviewer 1 ·

Basic reporting

no comment

Experimental design

The author's research is very interesting, but it seems that I can't see any difference in clinical results or scientific researches between the method proposed by the author and the current method of assembling genome, that is, this change in methodology is not enough to bring practical benefits
The author advocates combining multiple repeat detection methods to improve the accuracy of identifying repeat sequences, but does not consider the time-consuming problem and the running memory requirements of the computer, which may affect the application scope of this method. This should be described or tested.

Validity of the findings

no comment

Additional comments

The author needs to discuss the content of the conclusion carefully and deeply.

·

Basic reporting

1. This article reviewed current methods for improving genome assembly with prior repeat identification or hybrid assembly approach and their data structures. The topic is of broad interest and has not been reviewed recently. Overall the paper is well organized logically, although the language can be further improved for better understanding of the context, as I noticed some grammatic errors in the article (line 79, 112-114, 139, 174-177, etc.).

2. In the Introduction section, authors mentioned that third generation sequencing has high error rate and sequencing cost, so NGS is the only accurate and affordable solution for genome sequencing (line 121-125). This is not really true, as the field of genome assembly using TGS data is growing rapidly recently. Many assemblers have been developed for TGS data (such as Canu, Flye, wtdbg2, hifiasm, Shasta), which can handle the higher error rate and achieve much higher continuity than NGS assembly. It is understandable that TGS assembly is not the focus of this review article. I would suggest authors to revise this section to avoid incorrect statements.

Experimental design

1. In the section 'Prior repeat identification methods' (starting line 251), all the evaluation was focused on the repeat identification accuracy, without considering its effect on the following assembly. As the ultimate goal of repeat identification is to improve genome assembly, a comparison of assembly quality/continuity for assemblers/tools without prior repeat identification and assemblers/tools with different repeat identification approaches can be helpful here.

2. In the section 'Accuracy and performance evaluation in genome assembly' (starting line 197), there are tools, such as QUAST, Merqury, and Inspector, that can evaluate the accuracy of assembly, instead of just evaluating continuity (N50). Assembly errors within each contig can be identified and then used to calculate base accuracy of the assembly, which can be more informative than N50. I would suggest to add these methods into this section.

3. The referenced articles are not cited in ascending order. For example, reference [45], [14], and [47] occurred before [1] (line 55-59).

Validity of the findings

Based on the results and discussion in the article, I don't think the conclusions are strongly supported.

1. One major conclusion in this article is that 'as a result of this review, most prior repeat identification methods can detect only a specific length of repeat' (line 37-38). However, no results were shown regarding the relationship between repeat identification accuracy and the size of repeats. I would suggest to benchmark different repeat identification methods on a simulation dataset containing repeats of different sizes, so that the accuracy can be directly assessed at different sizes to support the conclusion.

2. The methods of each tool were explained clearly with the help of figures. However, more comparison between different tools can be helpful for readers to understand the difference in mythology and the strength/limitations of each tool. A table listing salient features for each tool, or a merged figure containing all tools may be helpful.

3. In Table 1, the authors claimed that SWA outperformed other tools as it revealed higher repeat size and count (line 288-290). It is not true since SGA reported higher repeat count than SWA. In addition, SWA achieved lowest N50 among all tested tools, suggesting a worst assembly continuity for SWA. This seems to be conflict with authors' statement that prior repeat identification can improve assembly quality. The authors did not comment on the N50 - do they have any explanation for why this might be?

4. In Table 3, the performance of 'with DACCOR' and 'without DACCOR' was evaluated on two distinct samples AR1 and AR2. I would suggest to do an unbiased comparison in the same sample (or using both samples), as there might be natural divergence between two datasets.

Additional comments

Some minor points for clarification:

Line 104: an extra parenthesis.
Line382: Should be 'III' instead of 'II'

---

## Round 0.2 · Minor Revisions

I was totally satisfied with the changes introduced as a reaction to the reviewers' comments. However, you (post-Accept!) introduced so many NEW edits that this will require further review. Please, prepare a new carefully formatted version and then supply your revised manuscript.

This resubmission will need to be checked. Don't worry - it is mainly a technical issue.

·

Basic reporting

No comment.

Experimental design

No comment.

Validity of the findings

No comment.

---

## Round 0.3 · accepted · Accept

Thank you for carefully handling the changes in the newest version.